# Safety and Immunogenicity of the GamTBvac, the Recombinant Subunit Tuberculosis Vaccine Candidate: A Phase II, Multi-Center, Double-Blind, Randomized, Placebo-Controlled Study

**DOI:** 10.3390/vaccines8040652

**Published:** 2020-11-03

**Authors:** Artem P. Tkachuk, Evgeniia N. Bykonia, Liubov I. Popova, Denis A. Kleymenov, Maria A. Semashko, Vladimir P. Chulanov, Sergey B. Fitilev, Semyon L. Maksimov, Elena A. Smolyarchuk, Victor A. Manuylov, Daria V. Vasina, Vladimir A. Gushchin, Alexander L. Gintsburg

**Affiliations:** 1N.F. Gamaleya Federal Research Centre for Epidemiology and Microbiology, Ministry of Health of the Russian Federation, 123098 Moscow, Russia; evgeniya_bikonya@mail.ru (E.N.B.); ljubovprokudina@gmail.com (L.I.P.); 10000let@rambler.ru (D.A.K.); maria.a.semashko@gmail.com (M.A.S.); victormanuilov@yandex.com (V.A.M.); d.v.vasina@gmail.com (D.V.V.); gintsburg@gamaleya.org (A.L.G.); 2National Medical Research Center for Physiopulmonology and Infectious Diseases, Ministry of Health of the Russian Federation, 127473 Moscow, Russia; vladimir@chulanov.ru; 3The Federal State Autonomous Educational Institution of Higher Education I.M. Sechenov First Moscow State Medical University of Ministry of Healthcare of the Russian Federation (Sechenovskiy University), 119991 Moscow, Russia; smolyarchuk@mail.ru; 4Department of General and Clinical Pharmacology, RUDN University, 117198 Moscow, Russia; fitilevsb@yandex.ru; 5City Polyclinic No. 2, Moscow Department of Health, 117556 Moscow, Russia; 6Federal State Budgetary Educational Institution of Higher Education A.I. Yevdokimov Moscow State University of Medicine and Dentistry of the Ministry of Healthcare of the Russian Federation, 127473 Moscow, Russia; maximov_s@bk.ru; 7Infectious Clinical Hospital No. 2, 117556 Moscow, Russia; 8Department of Virology, Lomonosov Moscow State University, 119991 Moscow, Russia

**Keywords:** tuberculosis, subunit vaccine, BCG booster, clinical trials, safety and immunogenicity

## Abstract

GamTBvac is a candidate tuberculosis vaccine with two fusion proteins, containing Ag85a, ESAT6, CFP10, and a dextran-binding domain (DBD). Phase II of a double-blind, randomized, multicenter, placebo-controlled study in parallel groups in healthy adults to evaluate the safety and immunogenicity of GamTBvac in 180 previously-vaccinated with Bacillus Calmette–Guérin vaccine (BCG) healthy volunteers without *Mycobacterium tuberculosis* (MTB) infection was conducted. The dose (0.5 mL) of either the study drug or a placebo was administered subcutaneously twice with an 8-week interval. At eight timepoints from 14 to 150 days, whole blood and sera were assayed. Antigen-specific T-cell responses were measured by an in-house interferon-gamma release assay (IGRA-test), the QuantiFERON (QTF) test, and intracellular cytokine staining (ICS). For antibody response detection, the bead-based multiplex immunoassay (MIA) was applied. The vaccine confirmed an acceptable safety profile previously shown in a first-in-human clinical study. After stimulation with both fusions, the highest median level of INF-γ was detected on day 21. The GamTBvac vaccine induced antigen-specific interferon-gamma release, Th1 cytokine-expressing CD4^+^ T-cells, and IgG responses and results support further clinical testing of GamTBvac.

## 1. Introduction

Tuberculosis (TB) remains one of the world’s most serious infectious diseases, causing about 1.4 million deaths annually [1]. Globally, an estimated 10.0 million people fell ill with TB in 2018. Vaccination of children with the Bacille Calmette–Guérin (BCG), treatment of people with active and latent TB infections (1.7 billion people already infected), and prevention of transmission of *Mycobacterium tuberculosis* (MTB) through early detection using the QuantiFERON test and the purified protein derivative test (PPD skin test) are the main anti-TB activities worldwide [1]. The BCG vaccine exhibits effective protection for newborns from miliary tuberculosis and tuberculous meningitis. However, numerous efficacy studies conclude that the BCG vaccination in infants does not prevent infection with MTB or the reactivation of latent tuberculosis in adults. Thus, BCG vaccination does not prevent tuberculosis transmission [2].

To reach The End TB Strategy targets set for 2030 and 2035 by the World Health Organization (WHO), technological breakthroughs are needed to decrease the global TB incidence rate by 2025. One of the main priorities of the Strategy is the development of novel prophylactic and therapeutic vaccines, rapid diagnosis, and more simple, effective regimens for treating TB disease [1].

The most promising types of candidate vaccines against TB are recombinant modified BCG vaccines, attenuated strains of *M. tuberculosis*, recombinant subunit vaccines, and DNA vaccines. Currently, 14 candidate TB vaccines are undergoing clinical trials aimed at replacing BCG or at boosting its immunity [3,4,5,6,7].

Recently, we reported the first-in-human trial of the BCG booster candidate vaccine GamTBvac [8]. GamTBvac is a recombinant vaccine containing three MTB antigens: the early secreted antigenic target 6 kDa (ESAT6), the 10-kDa culture filtrate protein (CFP10) and Ag85a, fused into two chimeric proteins with a dextran-binding domain (DBD) from *Leuconostoc mesenteroides*. These fusions were formulated with the adjuvant containing dextran 500 kDa, diethylaminoethyl (DEAE)-dextran 500 kDa, and CpG oligodeoxynucleotides (ODN). Sixty MTB-uninfected, BCG-vaccinated, healthy volunteers participated in that study. Three different doses of the vaccine with a double-immunization scheme were evaluated for immunogenicity and safety. The vaccine was safe and well-tolerated and induced a significant interferon gamma (IFN-γ) increase in the in-house interferon-gamma release assay (IGRA-test) and specific IgG response in a multiplex bead-based analysis. It was shown that a half dose causes a stable and high immune response to all antigenic components of the vaccine [8]. Here we present the results of a Phase II, double-blind, randomized, placebo-controlled trial to evaluate the immunogenicity, safety, and reactogenicity of GamTBvac in 180 BCG-vaccinated healthy volunteers.

## 2. Materials and Methods

### 2.1. Vaccine Production

Vaccine composition and production are described in detail in the patent RU 2 665 817 C1 and in our previous publications on GamTBvac preclinical and clinical studies [8,9]. The vaccine was manufactured by the Gamaleya Federal Research Centre for Epidemiology and Microbiology in Good Manufacturing Practice (GMP) accredited facility and supplied to the study site as a lyophilized product.

### 2.2. Study Design and Ethical Considerations

The GamTBvac Phase II double-blind, randomized, multicenter, placebo-controlled study in parallel groups in healthy BCG-vaccinated adults was conducted at 2 sites in Russia. The vaccine was administered twice with an 8-week interval (on day 0 and day 57), and the participants were observed for 5 months (150 days) after the first administration (Appendix A). The study was conducted in accordance with the Helsinki Declaration and Good Clinical Practices (ICH-GCP) and the legislation of the Russian Federation, externally monitored by an independently contracted research organization (Chromos Ltd., London, UK). The study was approved by the Council of Ethics at the Ministry of Health of the Russian Federation (Protocol No. 180 from 6 November 2018, permission of the Ministry of Health of the Russian Federation No. 571 from 14 November 2018) and local Ethics Committees of each research center (The Ethics Committee of the City Polyclinic No. 2, Moscow Department of Health, Moscow, Russia: Protocol No. 11 from 30 November 2018, The Local Ethics Committee of the Federal State Autonomous Educational Institution of Higher Education I.M. Sechenov First Moscow State Medical University of Ministry of Healthcare of the Russian Federation: Protocol No. 11–18 of from 5 December 2018). Written informed consent was obtained from all participants. This trial was registered on the clinical trial database ClinicalTrials.gov with ID NCT03878004.

### 2.3. Recruitment and Enrolment

Adult healthy volunteers from 18 to 49 years of age, both sexes, HIV-negative, generally healthy based on medical assessment, and who were BCG-immunized were enrolled for the study. Prior BCG immunization was confirmed by a record in the volunteers’ medical history or by the presence of a characteristic scar. Volunteers had a positive reaction (5–9 mm) in response to tuberculin (PPD test). Participants were *Mycobacterium tuberculosis* uninfected as defined by a negative QuantiFERON TB Gold ELISA test (Qiagen, Germantown, MD, USA) and Diaskintest (Generium, Moscow, Russia).

Screening procedures included the collection of personal medical history, a physical examination, chest radiography, an electrocardiogram (ECG), blood collection for baseline chemistry and hematology analysis, general urine analysis, and Hepatitis B and C serology. No pregnant or lactating women were included (Appendix A).

Volunteers who met the inclusion criteria were randomly allocated to the GamTBvac vaccine group or to the placebo group in a 3:1 ratio. The dose (0.5 mL) of either a study vaccine or a placebo containing Dextran 500 kDa and DEAE-Dextran 500 kDa (Pharmaceutical Quality, Pharmacosmos, Denmark) was administered subcutaneously. The administration of the study drug or placebo was performed twice with an 8-week interval.

### 2.4. Follow-Up and Safety Evaluation (Clinical Procedures)

After each vaccination on day 0 and day 57, volunteers were admitted to the hospital for 24 h for continuous monitoring. The physical examination was performed on days 0 (before and after vaccination), 1, 7, 21, 30, 57, 58, 64, 78, 87, 120, and 150. Blood for biochemistry and hematology tests, urine for urinalysis, and electrocardiogram (ECG) data were collected on days 0 (before vaccination), 1, 7, 30, 57, 58, 64, 87, and 150. A QuantiFERON (QTF) test was performed twice during the study: day 0 and day 150. Adverse events (AEs) were recorded during the entire study period, and their relatedness to the study drug was evaluated by investigators. Participants had to complete the patient diary card for 7 days after each vaccination to evaluate solicited AEs.

### 2.5. Immunogenicity Assays

Venous blood samples were collected to evaluate immunogenicity before vaccination at the screening visit, as well as on day 0 immediately after the first vaccination, and then on days 21, 57 (immediately after the second vaccination), 64, 78, 87, 120, and 150. In all immunogenicity assays except the QTF test, 171 volunteers were analyzed, among them, 129 were from the GamTBvac vaccine group, and 42 were from the placebo group. The remaining 8 participants were excluded from the immunoassays, as they missed several timepoints of blood sample collection for valid reasons.

The following immunogenicity parameters were evaluated: specific IFN-γ secretion level in response to stimulation with vaccine antigens, the percentage of CD4^+^/CD8^+^ Τ cells expressing IFN-γ, TNF-α, and/or IL-2 in response to stimulation with vaccine antigens as well as IgG antibody responses to the fused vaccine antigens or their individual domains, determined using multiplex suspension fluorescent-linked immunosorbent assay.

#### 2.5.1. Cell-Mediated Immune Response

*Whole blood intracellular cytokine staining assay (ICS)*. Vaccine-specific CD4^+^ and CD8^+^ T cell responses were evaluated by whole-blood intracellular cytokine staining and flow cytometry on days 0, 57, 87, 120, and 150. The protocol was described previously and used with modifications [10]. Briefly, each whole-blood sample (500 µL) was stimulated with vaccine antigens: either DBD-ESAT6-CFP10 fusion (fDEC) protein (10 µg/mL) or DBD-Ag85a fusion (fDA) protein (10 µg/mL). An unstimulated control (no antigen) and positive control (phytohemagglutinin, 7.5 µg/mL; PanEcko, Russia) were included in each assay. Costimulatory antibodies anti-CD28 and anti-CD49d at 0.45 µg/mL (BD Biosciences, Franklin Lakes, NJ, USA) were added to all test tubes. Whole-blood samples were incubated for 5 h at 37 °C and 5% CO_2_ in an incubator and then treated with 10 µg/mL Brefeldin-A (Sigma–Aldrich, St. Louis, MO, USA), followed by an additional overnight incubation not exceeding 16 h under the same conditions. Subsequently, blood samples were stained with LIVE/DEAD aqua viability dye (Life Technologies, Carlsbad, CA, USA), anti-human fluorochrome-conjugated antibodies to surface markers anti-CD3-PerCP-Cy5.5 (BD Biosciences, Franklin Lakes, NJ, USA), and anti-CD4-BV785 (BioLegend, San Diego, CA, USA), anti-CD8-BV650 (BioLegend, San Diego, CA, USA). Afterward, red blood cells were lysed using home-buffer [11], then the remaining white cells were washed twice with phosphate-buffered saline (PBS buffer) (VWR Life Science, Radnor, PA, USA), fixed, and permeabilized using IntraPrep Permeabilization Reagent (Beckman Coulter, Brea, CA, USA). Subsequently, cells were washed again and stained with anti-human antibodies to intracellular markers anti-IFN-γ-FITC (Beckman Coulter, Brea, CA, USA), anti-IL-2-PE (Beckman Coulter, Brea, CA, USA), and anti-TNF-α-Alexa Flour 700 (BioLegend, San Diego, CA, USA). After that, the white cells were washed twice with PBS buffer.

Cells were analyzed by a Cytoflex LX flow cytometer (Beckman Coulter, Brea, CA, USA). Analysis of the flow cytometry data was performed using CytExpert software version 2.3, and 1,000,000–1,500,000 events per sample were recorded.

The following gating strategy was used. Lymphocytes were gated with forward scatter area (FSC-A) versus side scatter area (SSC-A), and the singlets were selected by plotting FSC-A versus forward scatter height (FSC-H). CD3-positive cells were gated by plotting CD3-PerCP-Cy5.5 versus SSC-A. Boolean gating was used for the detection of cells producing any combination of cytokine (IFN-γ, IL-2, TNF-α). Each variant (single, double, and triple-positives) was expressed in the percentage of the total CD4^+^ or CD8^+^ T cells. The background (unstimulated control) was subtracted from all received values.

*Interferon-gamma release assay (IGRA-test).* An in-house interferon-gamma release assay test (IGRA) was performed on days 0, 21, 57, 78, 87, 120, and 150. The method was performed as previously described with some modifications [8]. Briefly, under sterile conditions, 50 µL of whole blood was supplied with 300 µL of growth medium (90% RPMI-1640 medium, 10% fetal bovine serum, 2 mM L-glutamine, 10 mM HEPES, 50 µg/mL gentamicin sulfate (PanEco, Moscow, Russia). Samples were stimulated with either vaccine antigens DBD-Ag85a (50 µg/mL) or DBD-ESAT6-CFP10 (50 µg/mL) or the non-specific stimulator Concanavalin A (4 µg/mL, positive control; Sigma–Aldrich, Taufkirchen, Germany). Blood samples without any stimulation were used as a negative control. Samples were incubated for 72 h at 37 °C and 5% CO_2_ in an incubator. Afterward, released IFN-γ was quantified using an IFN-ELISA-BEST kit (A-8752, Vector-Best, Novosibirsk, Russia). Antigen-specific response was determined as the difference between measured values of the stimulated probe and negative control.

#### 2.5.2. Humoral Immune Response

*Multiplex immunoassay (MIA).* Antibody IgG responses specific to GamTBvac vaccine subunits (Ag85a, ESAT6, CFP10, DBD, and their fusions, DBD-Ag85a and DBD-ESAT6-CFP10) were determined by a bead-based multiplex assay using xMAP technology (Luminex Corporation, Northbrook, IL, USA). Coupling antigens to microspheres (5–20 µg per 10^6^ microspheres) was performed according to previous protocols [12,13]. The assay was described previously [8] and accomplished with some modifications. In brief, 80 µL of CBS-T buffer (PBS, 1% BSA, 0.1% Tween-20, 0.05% NaN_3_) with 2500 microspheres per region and 20 µL of serum 20-fold prediluted with PVXC buffer (PBS, 0.8% polyvinylpyrrolidone, 0.1% casein, 0.5% polyvinyl alcohol, 0.05% NaN_3_) were incubated for 60 min at 25 °C and 800 rpm in the microplates (MICROLON 96 W Microplate, Greiner, Nussbach, Austria).

Then, microplates were washed with PBS-TBN buffer (PBS, 0.1% BSA, 0.03% Tween-20, 0.05% NaN_3_) using the 405 TS Microplate Washer (BioTek, Winooski, VT, USA). Then, 100 µL of 2.5 µg/mL phycoerythrin conjugated anti-human IgG (OneLambda/ThermoFisher Scientific, Waltham, MA, USA) was added to the microspheres into the wells. Further incubation continued for 30 min at 25 °C and 800 rpm, followed by washing. Microplates were subsequently analyzed by the MAGPIX (Luminex, Northbrook, IL, USA). The presence of vaccine-specific antibodies IgG was detected by measuring the mean fluorescence intensity (MFI) at any time point.

#### 2.5.3. QTF Test

The QuantiFERON test was performed using a commercially available QuantiFERON-TB Gold ELISA reagent kit (QFT, Qiagen, Germantown, MD, USA) in accordance with the manufacturer’s instructions.

### 2.6. Statistical Analysis

Statistical analysis was performed using the R software package (www.r-project.org). All data were presented as descriptive statistics. Nominal data were represented by absolute and relative frequencies (percentage), and continuous data were represented by the number of missed observations, average value, standard deviation, median, quartiles, and minimum and maximum values.

Safety was assessed by the comparison of the frequency of AEs between the groups using Fisher’s exact test. A correction for multiple Hill comparisons was applied to the set of *p*-values obtained.

Statistical analysis of immunogenicity data was performed with GraphPad Prism 7 (San Diego, CA, USA). The dataset was analyzed with nonparametric tests. To compare differences between timepoints in one group/antigen combination, the Friedman test with multiplicity adjustment by Dunn’s comparison test was used. For the cross-referencing of GamTBvac recipients and the placebo group, we performed the Mann–Whitney test with Dunn’s post hoc test.

For the results of the QTF test, a comparison between two treatment groups was performed using the Mann–Whitney test.

## 3. Results

### 3.1. Participants

From 13 December 2018 to 15 November 2019, 236 volunteers were included in the study. During the screening period, 56 of them were excluded due to noncompliance with the inclusion criteria. Among 180 enrolled participants, 135 were randomized to the GamTBvac vaccine group and 45 to the placebo group (Appendix A).

Overall, 179 volunteers received at least one dose of the study vaccine and had at least one immunogenicity evaluation: 134 participants were allocated to the GamTBvac group, and 45 participants were allocated to the placebo group; 176 volunteers completed the treatment protocol. The population included in the immunogenicity analysis is described in the section Immunogenicity Assays (Section 2.5). The population for the safety assessment included all 180 participants. There was no statistically significant difference by gender (*p* = 0.388), nor by age (*p* = 0.411), nor by body mass index (BMI) (*p* = 0.869) between the groups. Demographic characteristics of the enrolled participants who received at least one dose of the investigational product and had at least one immunogenicity evaluation are shown in Appendix A.

### 3.2. Safety

AEs were recorded throughout the study, from the first vaccine administration to 150 days after the first vaccination. During the study period, there were no serious adverse events (SAEs), deaths, or other significant events. The total number of AEs was significantly higher (*p* < 0.001) in the vaccine group than in the placebo group: 426 AEs (in 90.4% of participants) in the GamTBvac group compared to 74 AEs (in 53.3% of participants) in the placebo group (Appendix A).

The majority of AEs were mild or moderate. Only one (0.7%) participant receiving the study drug had a severe AE—an increase in aspartate aminotransferase level up to 686 U/mL with the subsequent alanine aminotransferase level increase. The AE occurred on day 30, did not require the patient’s withdrawal from the study, and resolved in 2 weeks.

In the GamTBvac group, 350 (83.7%) cases were considered as being related to vaccination, 50 (21.5%) were considered as not related, and for 26 (11.9%) cases, it was not possible to define the relatedness. In the placebo group, 37 (40.0%) AEs were related to the study, 27 (24.4%) were considered as not being related, and for 9 (20.0%) cases, it was not possible to define the relatedness.

The most common AEs in the GamTBvac group (in both total and related to investigational product AEs) were the reaction at the injection site in 47.4% of volunteers, the pain at the injection site in 21.5% of volunteers, and the C-reactive protein level increase in 24.4% of volunteers. In the placebo group, the frequency of these AEs was 4.4% for the reaction at the injection site, 2.2% for the pain at the injection site, and 15.6% for the C-reactive protein level increase.

The immediate type of AEs developed within 2 h postvaccination in 1.5% of volunteers receiving the vaccine and in 2.2% volunteer receiving the placebo.

Solicited and unsolicited AEs developed within 7 days postvaccination were studied separately. The list of solicited AEs (Table 1) included both local reactions (such as pain, redness, hematoma, edema, induration at the vaccination site) and systemic reactions (fever, myalgia, general malaise).

There were, in total, 219 solicited AEs (in 74.1% of volunteers) in the GamTBvac group, but only 1 AE (vaccination site reaction) had moderate intensity, and the rest were mild. A total of 212 AEs (in 74.1% of volunteers) were related to the study vaccine, 1 (0.7%) was not related, and for 6 AEs (in 3.7% of volunteers), relatedness remained unclassified. The majority of solicited AEs (212 AEs in 73.3% of volunteers) were local reactions to vaccine administration. Systemic reactions included increased body temperature in 3.0% of volunteers, myalgia in 2.2% of volunteers, and three cases of malaise in 2.2% of volunteers.

In the placebo group, there were nine solicited AEs (in 15.6% of volunteers). All AEs had mild intensity. There were eight AEs (13.3%) related to the study drug, and one AE (2.2%) remained unclassified. The majority of solicited AEs (five AEs in 8.9% of volunteers) were local reactions to vaccine administration. Systemic reactions included increased body temperature in 2.2% of volunteers and myalgia in 4.4% of volunteers (Table 1). There were no solicited AEs registered later than 7 days postvaccination.

The most frequent unsolicited AEs were registered within 7 days postvaccination based on laboratory and instrumental data in both groups. In the GamTBvac group, there were 98 unsolicited AEs in 40.0% of volunteers. Of these, 89 AEs (39.3%) were mild, and 9 AEs (3.7%) were moderate. A total of 81 AEs (34.1%) were related to the study drug, 5 AEs (3.7%) were not related, and for 12 AEs (5.2%), relatedness remained unclassified. In the placebo group, there were 32 unsolicited AEs (31.1%). Only one AE (2.2%) was moderate, and the rest were mild. A total of 19 AEs (20.0%) were related to the study drug, 9 AEs (6.7%) were not related, and 4 AEs (8.9%), relatedness remained unclassified (Table 2).

In the GamTBvac group, the results of the hematology and biochemistry blood tests were normal or had clinically insignificant deviations in most cases. Clinically significant deviations below normal values were observed in the following parameters: hemoglobin, hematocrit, red blood cells, platelets, white blood cells, segmented neutrophils and lymphocytes, urea, glucose, and calcium. In the placebo group, the majority of deviations in hematology and biochemistry parameters were clinically significant as well.

### 3.3. Immunogenicity

#### 3.3.1. Antigen-Specific INF-γ Response

To detect cell-mediated immunogenicity of GamTBvac for recipients and the placebo group, an in-house IGRA-test was conducted, as described previously [11]. Levels of INF-γ production after blood stimulation with recombinant antigens were determined at all study time points.

After stimulation with DBD-Ag85a on day 21, the highest median level of INF-γ was detected when compared to the baseline (*p* < 0.0001) (Figure 1). A slight drop in INF-γ expression was noted on day 57, and after the second immunization, the median cytokine level remained nearly the same.

Close dynamics of INF-γ production were revealed after stimulation with DBD-ESAT6-CFP10 (Figure 1). However, the median cytokine level was several times lower, compared to DBD-Ag85a stimulation (*p* < 0.0001 at any post-vaccination time point).

Based on these results, the following criteria for subjects responding to GamTBvac vaccination were estimated: First, the baseline level of INF-γ production was lower than by the end of the study (day 150); second, their cell-mediated immune responses persisted by day 150 and exceeded the 95th percentile of the pre-vaccination level (*p* < 0.05). Accordingly, there was a 41% response rate among the vaccinated volunteers after stimulation with DBD-ESAT6-CFP10 and 52% with DBD-Ag85a.

Statistically significant differences (*p* < 0.01–0.0001) between vaccine recipients and the placebo group were noted from study day 21 (after the first immunization) until the end of the study (day 150, Figure 1). Thus, the increase in INF-γ production after the first immunization with GamTBvac correlates with previously described data [8], and we suggest it as a vaccine-induced immune response.

#### 3.3.2. QTF Test Evaluation

The results of the QuantiFERON test were evaluated in a population with at least one immunogenicity assessment that included 179 volunteers. The QTF test was performed at the screening visit and at the last visit on day 150. At the screening visit, the result of the QTF test was negative for all volunteers, both from the vaccination group and from the placebo group. At the visit on day 150, the result was negative in 122 volunteers (91.0%) from the GamTBvac group and in 44 volunteers (97.8%) from the placebo group. No statistically significant difference between the groups was observed.

#### 3.3.3. Antigen-Specific T-Cell Response

Whole blood ICS was performed for all participants on the days of immunizations—0 and 57, and after the second immunization—87, 120, and 150. Both recombinant antigens DBD-Ag85a and DBD-ESAT6-CFP10 were used for an in vitro stimulation of whole blood samples on behalf of the detection of CD4^+^ and CD8^+^ T-cell responses.

A comparison between the placebo and vaccine recipient’s group was estimated at all timepoints, despite the relatively low frequencies of Th1-responses detected. Among them, there was almost no significant difference in the quantity of single and triple cytokine-expressing T-cells (Figure 2, Appendix A).

Compared to the placebo, in the vaccine recipient’s group, after the stimulation with DBD-Ag85a, the statistically significant increase in the value of cytokine production was detected for TNF-α^+^IL-2^+^CD4^+^ on study days 87, 120 (*p* < 0.0001), and 150 (*p* < 0.001) and for INF-γ^+^IL-2^+^CD4^+^ T-cells on days 87 (*p* < 0.01) and 120 (*p* < 0.05). After the stimulation with DBD-ESAT6-CFP10 such an increase was mainly noted for TNF-α^+^IL-2^+^CD4^+^ on study days 57 (*p* < 0.05), 87, 120, and 150 (*p* < 0.0001), and also to a much lesser extent for other double- and triple-positive specific CD4^+^ T-cells (Appendix A).

Cytokine-expression profiles of fusion-specific CD4^+^ T-cell responses (within the vaccine recipient’s group) were estimated. After the second immunization (day 57), we found an increase in the magnitude of the responses to DBD-Ag85a as well as to DBD-ESAT6-CFP10 fusions (Figure 2).

For DBD-Ag85a fusion, the production of specific INF-γ^+^ was detected from day 57. On study day 87, responses were detected in populations of IL-2^+^, INF-γ^+^, and TNF-α^+^ (single-positive), TNF-α^+^IL-2^+^ and INF-γ^+^TNF-α^+^ (double-positive), and TNF-α^+^IL-2^+^INF-γ^+^ T-cells. Only frequencies of specific bifunctional and trifunctional T-cells significantly increased compared to the baseline until day 150. Among the Th1-responses to DBD-ESAT6-CFP10 fusion, we registered a considerable increase in single-positive and TNF-α^+^IL-2^+^ and INF-γ^+^TNF-α^+^ double-positive and TNF-α^+^IL-2^+^INF-γ^+^ CD4^+^ T-cells on study day 57 and until the end of the study (Figure 2A,B).

Within the CD8^+^ T-cell responses, solely single-cytokine producing cells were detected, which correlates with previously described data about other sub-unit vaccines [10]. We observed only the expression of INF-γ producing cells from day 57 and by day 120, post first immunization under stimulation with both antigen fusions.

#### 3.3.4. Antigen-Specific IgG Response

The humoral immune response was studied in the sera of all volunteers at all studied time points, as well as in sera received 14 days before vaccination (point 0). The IgG levels of these sera samples did not differ significantly from the samples taken on the day of the first vaccination. For one person from the placebo group, an increase in the IgG antibodies level during the observation period was found, while for one volunteer from the vaccination group, there was no growth of IgG level.

In general, a significant increase in IgG levels in the vaccinated group was observed by the 21st day after the first vaccination in response to both fusions (anti-fDA IgG 24% and anti-fDEC 34% positives) and to three of their four components (with the exception of the ESAT6 antigen) (Figure 3, Appendix A). The peak of specific IgG levels was observed at 78 days after the first immunization, followed by a slight decrease towards the end of the study (*p* < 0.001). By day 78, 98% and 94% of volunteers from the vaccinated group showed an increase in the IgG antibody level to DBD-ESAT6-CFP10 and DBD-Ag85a, respectively.

We also examined all sera for the presence of specific IgG to each of the fusion components. Regarding fDA, DBD, and Ag85a, it was found that by the 21st day of the study among 31 volunteers who responded positively to fDA, one third (*n* = 11) were positive for antibodies to DBD only, and another third (*n* = 10) were negative for both DBD and to Ag85a (fDA(+) only group) (Figure 4A). On the 78th day of the study, among 121 fDA-positive volunteers, the proportion of exclusively DBD-positive volunteers increased to almost half (*n* = 54), while the number of those who did not answer either of the two components of this fusion was only four. The group of those positive for both DBD and Ag85a antigens (*n* = 59) also significantly increased, compared to the 21st day: 46 against 4%.

For fDEC, in a similar way, we can observe a slightly different situation (Figure 4B). On the 21st day of the study, among 44 volunteers positive for the IgG to this fusion, a third of them (fDEC(+) only group, *n* = 15) did not respond to any of its components, half were positive either simultaneously for DBD and CFP10, or only for CFP10, and only 6 people were positive for antibodies to DBD. In addition, at this time point, antibodies to ESAT6 were not detected in any person. By day 78, among 126 IgG–fDEC(+) volunteers, a small group showed a response only to CFP10 (*n* = 10), a fifth of them (*n* = 27) responded to all three components, and the vast majority (*n* = 87) were IgG-positive to DBD and CFP10 both (Figure 4B). In general, a moderate role in the formation of antibodies by the Ag85a component, and an extremely modest contribution to this process by ESAT6 should be noted.

## 4. Discussion

In a Phase II clinical study, the subunit recombinant tuberculosis vaccine GamTBvac was investigated for safety and immunogenicity. It confirmed an acceptable safety profile on a larger cohort of volunteers, previously shown in a first-in-human clinical study. No serious AEs or deaths were reported, and the majority of AEs had mild intensities. The most frequent AEs were vaccination site reactions and C-reactive protein increases, as were already shown in the previous clinical trial [8].

The solicited AEs were registered within 7 days postvaccination and included local vaccination site reactions and systemic reactions specific for immunization. The amount of reported solicited AE cases in the vaccine group was the half of the total observed cases in this group of volunteers. The most common local AEs were pain and vaccination site reactions, and the most common systemic reactions were an increased body temperature, myalgia, and malaise. All observed AEs were expected and were described in the investigators’ brochure.

The safety profile of the vaccine GamTBvac is consistent with the results obtained from clinical studies of the other tuberculosis subunit recombinant vaccines in healthy adults without or with TB infection, where the most often solicited local injection site reactions and systemic flu-like symptoms had been observed [14,15,16,17,18,19].

One of the main criteria for the estimation of TB-vaccine immunogenicity is the development of vaccine-induced T-cell responses [20,21]. Among the wide spectrum of different immunological assays, the most widely used for TB-vaccine development are IFN-γ ELISPOT, intracellular cytokine staining (ICS), and in-house IFN-γ-release assays [22].

After antigen-specific stimulation, we detected T-cell responses comprising only bifunctional TNF-α^+^IL-2^+^ CD4^+^ T-cells in comparison with the placebo group. The population of central memory T-cells, expressing double cytokines, was described previously in the study of AERAS-402, where they showed a decrease in this population lower than the limit of detection by the end of the study [23]. Here we discovered maximum frequencies of the CD4^+^ T-cell population expressing TNF-α^+^IL-2^+^ on study day 120 and a decrease 3 months after the second immunization (day 150, Appendix A).

Within the GamTBvac recipient’s group, we observed statistically significant specific CD4^+^ polyfunctional responses (including more than two cytokines, especially TNF-α^+^INF-γ^+^ and TNF-α^+^IL-2^+^INF-γ^+^) after the second vaccination (Figure 2). Among Th1-responses, double-positive and triple-positive T-cell responses are considered to be the main responses to either the BCG antigen or other subunit TB-vaccines. GamTBvac and other novel TB-vaccine candidates (MVA85 A, H56:IC31, H1:IC31, M72/AS01 E) showed related response profiles: CD4^+^ responses were higher than CD8^+^ T-cell responses, except AERAS-402 where the main response detected were polyfunctional and INF-γ^+^ expressing CD8^+^ T-cells [24]. However, the direct comparison of detected vaccine-induced T-cell responses with the ICS method is highly problematic due to the high variability of the parameters used in different laboratories for this type of analysis.

We also used an in-house IFN-γ-release assay to measure antigen-specific levels of IFN-γ expressed by T-cells. After the stimulation with both protein fusions included in the GamTBvac formulation, we detected an increase in IFN-γ production after the first immunization, and there was a uniform response without a significant drop in IFN-γ levels until the end of the observation. The median IFN-γ response towards DBD-Ag85a fusion was several times higher than towards DBD-ESAT6-CFP10 fusion (Figure 1). All the GamTBvac recipients were BCG-vaccinated; thus, Ag85a response could be explained as a booster effect in response to the BCG antigen component. In addition, we observed IFN-γ expression by CD4^+^ and CD8^+^ T-cells by ICS. It is well known that among Th1-responses, T-cells producing IFN-γ are the main component in the establishment of protective immunity against MTB [21,25].

We compared GamTBvac recipient’s IFN-**γ** level at day 150 vs. pre-vaccination and defined responders’ rate, which consisted of around 50% for each of both fusions. In other studies, when evaluating the responder rate using various methods (ICS, ELISPOT), the value of this indicator ranges from 50 to 90%. However, this does not allow researchers to assess whether this is a sufficient level of immunogenicity [23,26,27]. True effectiveness research is the aim of the third phase of clinical trials. We consider the results we have obtained to be sufficient grounds for such further testing.

To date, there are no proven mechanisms for the effects of humoral immunity in protecting against human tuberculosis. However, recent in vitro studies as well as in vivo murine model studies still suggest that its role in protection is underestimated [28,29,30]. For example, a positive correlation was revealed between the titer of IgG antibodies specific to the mycobacterial capsular polysaccharide arabinomannan and the ability to opsonize MTB for phagocytosis by macrophages, enhance phagolysosome fusion, and inhibit intracellular growth of pathogens [28]. Protective antibody responses have been identified in individuals with high exposure to MTB but without active TB. However, the protective effect of antibodies was dependent on both immune complexes and CD4^+^ T cells [29]. Furthermore, it was shown that antibodies from people with LTBI were better in inhibition of intracellular growth of the pathogen than antibodies from patients with active TB. Some differences in antibody profiles, in particular, glycosylation status of the crystallizable fragment (Fc), were revealed [30]. In our study, we evaluated the dynamics of the level of specific IgG antibodies to vaccine fusions DBD-ESAT6-CFP10 and DBD-Ag85a, as well as to their components (DBD, ESAT6, CFP, Ag85a). In general, a tendency toward an increase in the level of antibodies to all recombinant antigens was determined, reaching a maximum by day 78 after vaccination and their gradual decrease by day 150. In total, 98% and 94% of volunteers from the vaccinated group responded to DBD-ESAT6-CFP10 and DBD-Ag85a, respectively. Based on the assumption that the tertiary structure of the recombinant fusion and its individual components have similar epitopes, we tried to assess which component is predominantly involved in the formation of antibodies to fusion. Of all DBD-Ag85a positive responses, the percentage of DBD responses reached 88%, and for Ag85a responses reached 49% on day 78. For DBD-ESAT6-CFP10, the positive response to the DBD and CFP10 components represented 89 and 96%, respectively. Thus, we assumed that in both cases, the DBD subunit made a significant contribution in humoral responses to fusions and can probably enhance B-cell response to vaccine antigens. Looking at the other MTB-specific components of fusions, an antibody response to Ag85a and ESAT6 was found in only 50% and 21% of the volunteers.

Another interesting observation is that in the early stages (day 21), about 10% of the volunteers were IgG-positive for fusions but IgG-negative for their components. It can be assumed that the formation of the humoral immune response to the vaccine can be contributed not only by the epitopes of the individual components that make up the fusions but also by the spatial epitopes present exclusively in fusion proteins.

The development of effective tuberculosis vaccines is complicated by the lack of reliable protection correlates for MTB. Thus, the efficacy of vaccine candidates in preclinical studies cannot with confidence to predict protection in humans. Therefore, only late-stage clinical trials can accurately verify the protection.

## 5. Conclusions

In summary, in a Phase II clinical trial on 180 healthy volunteers, the vaccine was well-tolerated and induced a specific and durable Th1 and humoral immune response. The protective efficacy against MTB will be evaluated on a large population during the Phase III study.

## Figures and Tables

**Figure 1 vaccines-08-00652-f001:**
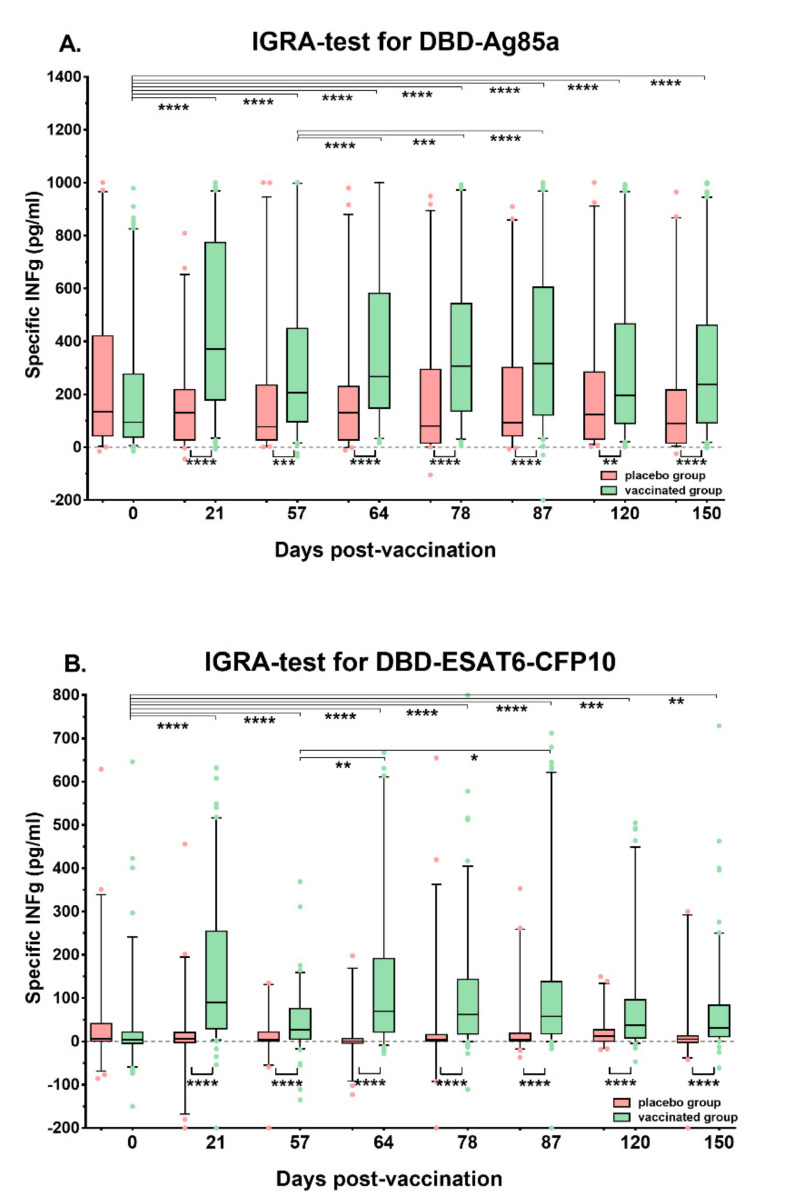
Dynamics of interferon gamma (INF-γ) production after whole-blood stimulation in volunteers and the placebo group, performed with an in-house INF-γ release assay (IGRA) on study days 0, 21, 57, 64, 78, 87, 120, and 150. (**A**). Stimulation of samples with DBD-Ag85a; (**B**). Stimulation of samples with DBD-ESAT6-CFP10; medians and interquartile ranges are shown. * *p* < 0.05, ** *p* < 0.01, *** *p* < 0.001, **** *p* < 0.0001.

**Figure 2 vaccines-08-00652-f002:**
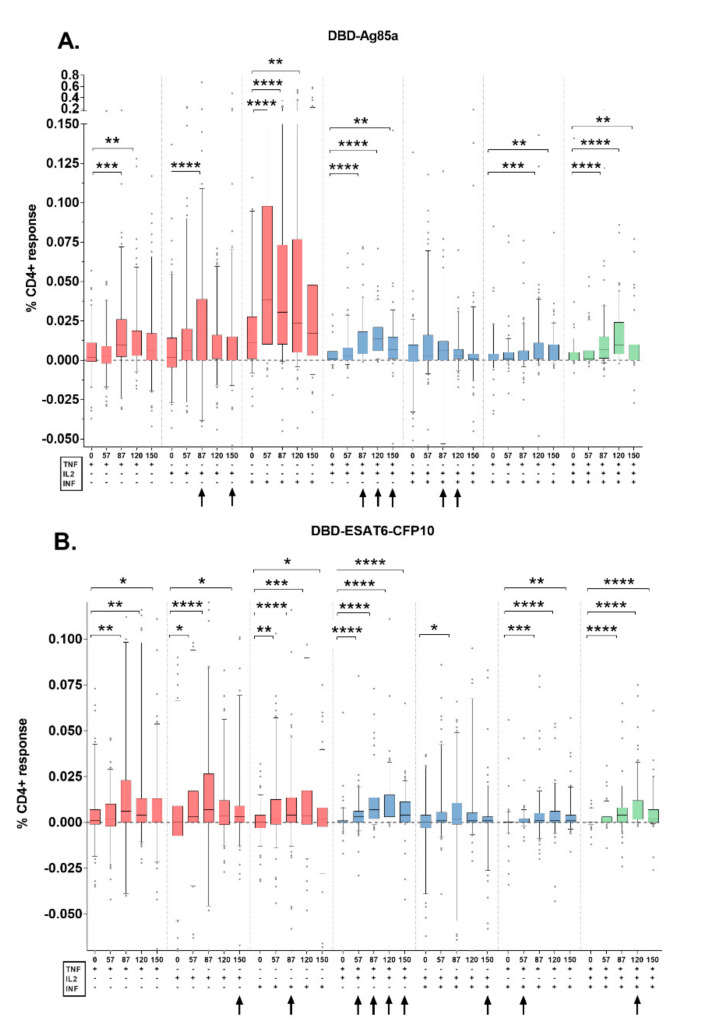
Cytokine expression profiles of CD4^+^ T-cells following stimulation with GamTBvac components in the vaccine recipients’ group. (**A**). Stimulation of samples with DBD-Ag85a; (**B**). Stimulation of samples with DBD-ESAT6-CFP10. Black arrows indicate specific CD4^+^ T-cells responses that differ significantly from those in the placebo group with a minimum *p* < 0.05. Medians and interquartile ranges are shown. * *p* < 0.05, ** *p* < 0.01, *** *p* < 0.001, **** *p* < 0.0001.

**Figure 3 vaccines-08-00652-f003:**
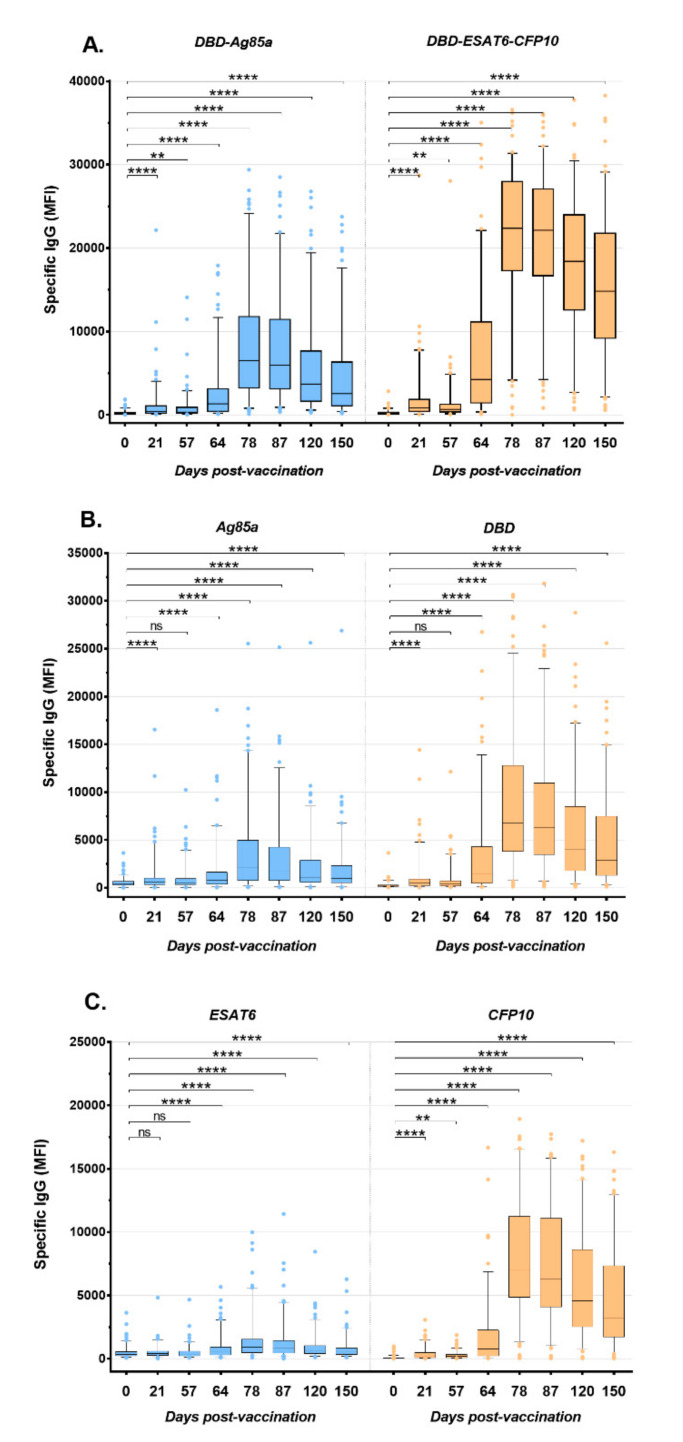
IgG antibody profiles after the immunization with GamTBvac specific to DBD-Ag85a and DBD-ESAT6-CFP10 (**A**), and individual vaccine components (**B**,**C**) in the group of vaccine recipients. ** *p* < 0.01, **** *p* < 0.0001, ns—not significant.

**Figure 4 vaccines-08-00652-f004:**
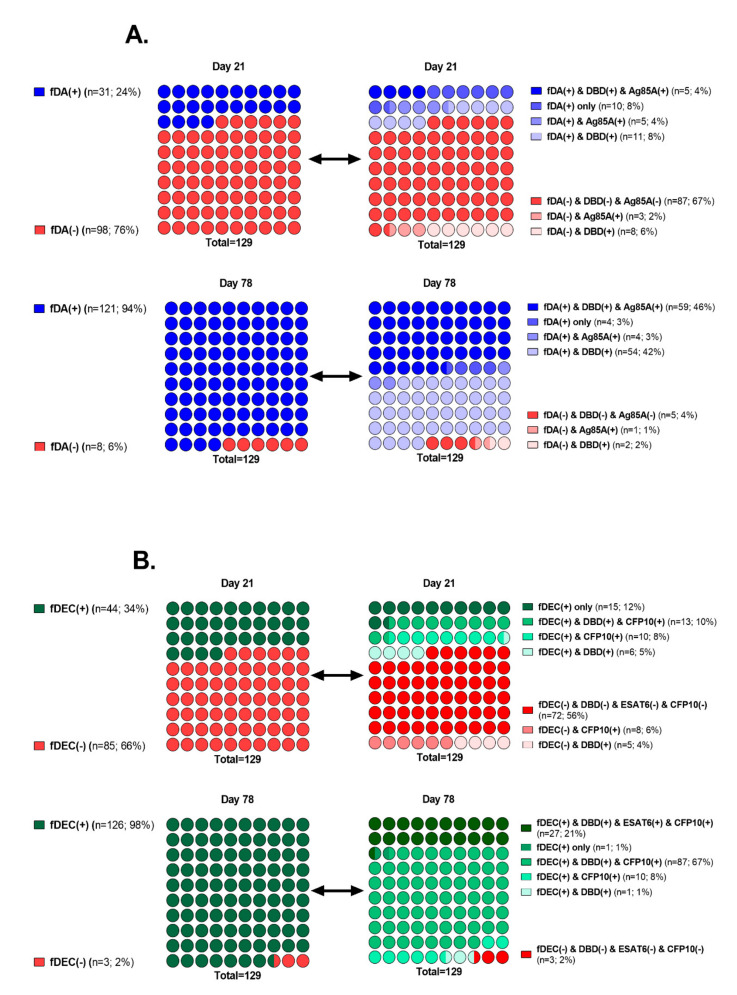
Humoral responses to vaccine fusions and their components in the immunized volunteers’ group on days 21 and 78. (**A**). Percentage ratio of positive/negative IgG responses to fusion DBD-Ag85a (fDA) is represented in the left dot plots. Percentage ratio of positive/negative IgG responses to fDA and its components is illustrated in the right dot plots. (**B**). Similar plots for fusion DBD-ESAT6-CFP10 (fDEC).

**Table 1 vaccines-08-00652-t001:** Solicited adverse events (AEs) and their intensity, safety evaluation set, *n* = 180. X = Number of volunteers with at least one AE; % = Percentage of volunteers with at least one AE in the group; Y = Total number of events. MedDRA (Medical Dictionary for Regulatory Activities) System Organ Class Term is written in bold.

MedDRA System Organ Class Term MedDRA Preferred Term	Intensity	GamTBvac *n* = 135		Placebo *n* = 45	
		X (%)	Y	X (%)	Y
Total number of volunteers with AEs	Mild	100 (74.1%)	218	7 (15.6%)	9
	Moderate	1 (0.7%)	1	0 (0.0%)	0
**General disorders and administration site conditions**	**Mild**	**99 (73.3%)**	**211**	**4 (8.9%)**	**5**
	**Moderate**	**1 (0.7%)**	**1**	**0 (0.0%)**	**0**
Injection site erythema	Mild	17 (12.6%)	20	1 (2.2%)	1
Injection site hematoma	Mild	4 (3.0%)	6	0 (0.0%)	0
Injection site induration	Mild	17 (12.6%)	21	1 (2.2%)	1
Injection site pain	Mild	29 (21.5%)	38	1 (2.2%)	1
Injection site reaction	Mild	5 (3.7%)	5	0 (0.0%)	0
Injection site swelling	Mild	7 (5.2%)	9	0 (0.0%)	0
Malaise	Mild	3 (2.2%)	3	0 (0.0%)	0
Vaccination site hematoma	Mild	2 (1.5%)	2	0 (0.0%)	0
Vaccination site reaction	Mild	64 (47.4%)	107	2 (4.4%)	2
	Moderate	1 (0.7%)	1	0 (0.0%)	0
**Investigations**	**Mild**	**4 (3.0%)**	**4**	**1 (2.2%)**	**2**
Body temperature increased	Mild	4 (3.0%)	4	1 (2.2%)	2
**Musculoskeletal and connective tissue disorders**	**Mild**	**3 (2.2%)**	**3**	**2 (4.4%)**	**2**
Myalgia	Mild	3 (2.2%)	3	2 (4.4%)	2

**Table 2 vaccines-08-00652-t002:** Unsolicited AEs and their intensity, safety evaluation set, *n* = 180. X = Number of volunteers with at least one AE; % = Percentage of volunteers with at least one AE in the group; Y = Total number of events. MedDRA (Medical Dictionary for Regulatory Activities) System Organ Class Term is written in bold.

MedDRA System Organ Class Term MedDRA Preferred Term	Intensity	GamTBvac *n* = 135		Placebo *n* = 45	
		X (%)	Y	X (%)	Y
Total number of volunteers with AEs	Mild	53 (39.3%)	89	14 (31.1%)	31
	Moderate	5 (3.7%)	9	1 (2.2%)	1
**Blood and lymphatic system disorders** Eosinophilia	**Mild**	**2 (1.5%)**	**3**	**1 (2.2%)**	**1**
	Mild	0 (0.0%)	0	1 (2.2%)	1
Leukocytosis	Mild	1 (0.7%)	1	0 (0.0%)	0
Neutrophilia	Mild	1 (0.7%)	1	0 (0.0%)	0
Reticulocytosis	Mild	1 (0.7%)	1	0 (0.0%)	0
**Gastrointestinal disorders**	**Mild**	**1 (0.7%)**	**2**	**0 (0.0%)**	**0**
Nausea	Mild	1 (0.7%)	2	0 (0.0%)	0
**General disorders and administration site conditions**	**Mild**	**3 (2.2%)**	**3**	**0 (0.0%)**	**0**
Asthenia	Mild	1 (0.7%)	1	0 (0.0%)	0
Hyperthermia	Mild	1 (0.7%)	1	0 (0.0%)	0
Tissue infiltration **Hepatobiliary disorders**	Mild	1 (0.7%)	1	0 (0.0%)	0
	**Mild**	**0 (0.0%)**	**0**	**1 (2.2%)**	**1**
Hyperbilirubinaemia	Mild	0 (0.0%)	0	1 (2.2%)	1
**Infections and infestations**	**Mild**	**1 (0.7%)**	**1**	**1 (2.2%)**	**1**
	Moderate	0 (0.0%)	0	1 (2.2%)	1
Herpes virus infection	Mild	1 (0.7%)	1	0 (0.0%)	0
Nasopharyngitis	Moderate	0 (0.0%)	0	1 (2.2%)	1
Rhinitis	Mild	0 (0.0%)	0	1 (2.2%)	1
**Injury, poisoning, and procedural complications**	**Mild**	**3 (2.2%)**	**3**	**0 (0.0%)**	**0**
Vaccination complication	Mild	3 (2.2%)	3	0 (0.0%)	0
**Investigations**	**Mild**	**41 (30.4%)**	**66**	**13 (28.9%)**	**27**
	**Moderate**	**4 (3.0%)**	**8**	**0 (0.0%)**	**0**
Alanine aminotransferase increased	Mild	1 (0.7%)	1	1 (2.2%)	1
Aspartate aminotransferase increased	Moderate	1 (0.7%)	1	0 (0.0%)	0
Bilirubin conjugated increased	Mild	2 (1.5%)	2	1 (2.2%)	1
Blood bilirubin increased	Mild	8 (5.9%)	10	5 (11.1%)	6
C-reactive protein increased	Moderate	3 (2.2%)	4	0 (0.0%)	0
Eosinophil count increased	Mild	9 (6.7%)	11	4 (8.9%)	5
Eosinophil percentage increased	Moderate	2 (1.5%)	3	0 (0.0%)	0
Hemoglobin decreased	Mild	21 (15.6%)	22	3 (6.7%)	4
Lymphocyte count decreased	Mild	3 (2.2%)	3	3 (6.7%)	3
Lymphocyte percentage decreased	Mild	0 (0.0%)	0	1 (2.2%)	1
Monocyte count increased	Mild	1 (0.7%)	1	1 (2.2%)	1
Neutrophil count decreased	Mild	2 (1.5%)	2	1 (2.2%)	1
Neutrophil count increased	Mild	1 (0.7%)	1	0 (0.0%)	0
pH urine increased	Mild	1 (0.7%)	1	0 (0.0%)	0
Protein total increased	Mild	1 (0.7%)	1	0 (0.0%)	0
White blood cell count decreased	Mild	7 (5.2%)	8	1 (2.2%)	2
**Metabolism and nutrition disorders**	**Mild**	**2 (1.5%)**	**2**	**0 (0.0%)**	**0**
	**Moderate**	**1 (0.7%)**	**1**	**0 (0.0%)**	**0**
Hyperkalemia	Mild	2 (1.5%)	2	0 (0.0%)	0
	Moderate	1 (0.7%)	1	0 (0.0%)	0
**Musculoskeletal and connective tissue disorders**	**Mild**	**3 (2.2%)**	**3**	**0 (0.0%)**	**0**
Myosclerosis	Mild	3 (2.2%)	3	0 (0.0%)	0
**Nervous system disorders**	**Mild**	**3 (2.2%)**	**3**	**0 (0.0%)**	**0**
Headache	Mild	3 (2.2%)	3	0 (0.0%)	0
**Respiratory, thoracic, and mediastinal disorders**	**Mild**	**2 (1.5%)**	**2**	**1 (2.2%)**	**1**
Cough	Mild	1 (0.7%)	1	0 (0.0%)	0
Respiratory disorder	Mild	1 (0.7%)	1	1 (2.2%)	1
Vascular disorders	Mild	1 (0.7%)	1	0 (0.0%)	0
Hypotension	Mild	1 (0.7%)	1	0 (0.0%)	0

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
