# Peer review of "Safety and Immunogenicity of the GamTBvac, the Recombinant Subunit Tuberculosis Vaccine Candidate: A Phase II, Multi-Center, Double-Blind, Randomized, Placebo-Controlled Study"

_vaccines, 2020, doi:10.3390/vaccines8040652_

Round 1

Reviewer 1 Report

The presented manuscript by A Tkachuk et al. describes the “Safety and immunogenicity of the GamTBvac, the recombinant subunit tuberculosis vaccine candidate: a phase II, multi-center, double-blind, randomized, placebo-controlled study”. The study includes the data analysis of the 180 participating volunteers at the time of vaccination/placebo until 150 days post first dose of GamTBvac. The analysis is well conduced regarding the experimental work as well as statistical analysis. The communication of the results is of importance to the field.

I have  a few questions/comments that may improve the clarify of the manuscript:

Based on figure 4 (whole blood assay, IFN-g response), the authors define criteria for subjects responding to GamTBvac based on their IFN-g level at day 150 vs pre-vaccination. Applying these criteria around 50% of the participants are classified as responders. Is this low proportion expected? Can this be explained in the discussion and what are conclusions for the future testing of GamTBvac?

Is the data presented in figure 5 and 6 taken from the above defined responder’s group or are all vaccinated individuals included? If only responders are presented, I would suggest to include a vaccinated vs non-vaccinated comparison as well to allow for a full comparison.

The note that the percentage of specific Th1 cells in figure 5 is very low. Was a positive control like PHA or a staining for Th1 transcription factor Tbet included in the experimental set up?

Reviewer 2 Report

The authors have reported the safety and immunogenicity results of a Phase II human trial of an experimental tuberculosis (TB) vaccine, GamTBvac, in 180 Russian volunteers. The manuscript is well-written. However, there are numerous methodological details missing, some of the graphs and tables contain redundant data or can be moved to the Supplement section or deleted, and there is too much unwarranted emphasis on the humoral response. The following questions and suggestions are meant to assist the authors to improve the clarity and impact of their manuscript.

            With respect to the Methods section, the following issues must be addressed by the authors.

  • Figure 1 is not correct. It shows that the volunteers were assigned to a treatment group BEFORE the screening process. That it not what it says in the text below (lines 115-118). Since there is no difference between the two groups, a single timeline will be sufficient and this revised figure can be moved to the Supplement section
  • The authors state (line 108) that the volunteers had a “,,,,,, positive reaction (5-9 mm)…..” to a PPD skin test. What does this mean? Why is this reaction size deemed “positive”? How can the authors be certain that this reactivity did not reflect cross-reactions with non-tuberculous mycobacteria (NTM)?
  • Please list the specific inclusion criteria used in this trial.
  • Please list the specific exclusion criteria used in this trial.
  • Were the volunteers screened for HIV? Were the volunteers screened for immunological reactivity to NTM?
  • The placebo group apparently did NOT receive the CpGODN that was contained in the vaccine. Why did the placebo group NOT receive the entire placebo material? This oversight may invalidate the “placebo controlled” aspect of the trial.
  • Adverse events (AE) were apparently “…….evaluated by investigators.” (line 126). This is not ethical. All adverse events must be investigated by an entirely independent Safety Committee to maintain the blinded aspect of the study and to ensure objectivity.

With respect to the Results section, the following suggestions should be seriously considered by the authors.

  • Figure 2 = Please delete. This is a standard method for evaluating intracellular cytokine production
  • Figure 3 (mis-labelled Figure 1) – Please move to the Supplement section. This distracts the reader from the main thrust of the paper.
  • Table 1 – Please move to the Supplement section. Since there are no differences between the vaccine and placebo groups, these data are not important enough to complicate the focus by the reader on significant results.
  • Table 4 – Please move to the Supplement section. This enormous table does not add to the authors’ description in the text and can be examined at the reader’s leisure.
  • Figure 4 – Panel C appears to contain the same data as Panels A & B; the data cannot be duplicated. Please remove Panel C.
  • Figure 4 – Why was there no boosting effect on IFNg production following the second dose of vaccine at Day 57. This does not bode well for this vaccine.
  • Please state which mycobacterial antigens are included in the QuantiFERON assay. Why was the QTF assay negative in more than 90% vaccine recipients at 150 days (line 128), while the laboratory-based assay was positive?
  • Figure 5 – This figure is difficult to visualize and interpret. There are very few significant responses – Why?
  • Figure 6 – Please delete this figure. It appears to contain exactly the same data as shown in Figure 5. The authors cannot present the same data twice.
  • Table 4 – Please state whether the percentages shown are statistically significantly higher than baseline.
  • Figure 8 – Please delete this figure. It appears to show essentially the same data as displayed in Figure 7. The authors cannot present the same data more than once.

With respect to the Discussion, the authors have placed far too much emphasis on the humoral responses.  Nearly half of the Discussion contains unwarranted speculation about the importance of the antibody responses in their vaccinees. There is little solid evidence that antibodies are involved in vaccine-induced resistance to TB.

Round 2

Reviewer 2 Report

No further comments

Author Response

No further comments needed.